# Molecular Dynamics Study of Laser Interaction with Nanoparticles in Liquids and Its Potential Application

**DOI:** 10.3390/nano12091524

**Published:** 2022-04-30

**Authors:** Hao Huang, Yingjie Xu, Guofu Luo, Zhuobin Xie, Wuyi Ming

**Affiliations:** 1School of Mechanical Science & Engineering, Huazhong University of Science and Technology, Wuhan 430074, China; huanghaomse@hust.edu.cn; 2Henan Key Lab of Intelligent Manufacturing of Mechanical Equipment, Zhengzhou University of Light Industry, Zhengzhou 450002, China; xyj15203873137@163.com; 3Guangdong Provincial Key Laboratory of Digital Manufacturing Equipment, Guangdong HUST Industrial Technology Research Institute, Dongguan 523808, China; xzb18336947581@163.com

**Keywords:** nanoparticles, laser interaction, nanobubble, molecular dynamics simulations, potential application

## Abstract

Laser interaction with nanoparticles in liquid is the fundamental theoretical basis for many applications but it is still challenging to observe this nanoscale phenomenon within a few nanoseconds in liquid by experiment. The successful implementation of the two-temperature method integrated with molecular dynamics (TTM-MD) in laser interaction with bulk material has shown great potential in providing a panoramic view of the laser interaction with the nanoparticles. However, the current TTM-MD model has to divide the system into cubic cells, which leads to mistakes near the nanoparticle’s surface. We introduce the latest model, which performs the TTM-MD on each individual cluster instead of the cubic cells, and its high-performance parallel cluster analysis algorithm to update the cluster size. The cluster-based TTM-MD revealed the nanoparticle formation mechanism of laser fragmentation in liquid (LFL) and facilitated the study of laser fluence’s effect on the size distribution. In addition to LFL, this model is promising to be implemented in the laser thermal therapy of tumors, laser melting in liquid (LML), etc. Although cluster-based TTM-MD has proven to be a powerful tool for studying laser interaction with nanoparticles, a few challenges and future developments for the cluster-based TTM-MD, especially the ionization induced by femtosecond, are also discussed.

## 1. Introduction

Nanoparticles (NPs) refer to particles with a size between 1 and 100 nm (also known as ultrafine particles), which belong to the category of colloidal particle size. They are distinguished from macroscopic objects in that their surface area accounts for a large proportion, and the surface atoms have neither long-order nor short-order amorphous layers. For example, when the particle diameter is 10 nm, the particle contains 4000 atoms. The surface atoms account for 40%. When the particle diameter is 1 nm, the particle contains 30 atoms, and the surface atoms account for 99%. It can be considered that the atoms’ state on the nanoparticle’s surface is closer to the gaseous state, while the atoms inside the particle may be in an ordered arrangement [1]. Hence, the fast growth of nanotechnology and nanoscience and its extensive applications in practically all industrial domains have occurred during the last decade [2]. In a similar vein, the urgent worldwide demand for different nanomaterials (NMs) has sped up the development of NPs synthesis technologies. For example, gold nanoparticles (GNPs) based on imaging, optical, and electrochemical sensing technologies, are used for diagnosis, treatment, and basic in vitro and in vivo toxicity of various diseases, especially cancer, Alzheimer’s disease, HIV, and other diseases, which widely used in the fields of medicine, chemistry, spectroscopy, biochemistry, biophysics and nanoscience [3].

Therefore, GNPs have many applications in different areas, from cancer phototherapy and diagnosis to targeted drug delivery [4,5,6,7,8,9,10]. Jazayeri et al. [9] confirmed that bladder cancer could be detected early by adopting anti-survivin antibody-conjugated GNPs, as depicted in Figure 1. In terms of COVID-19 antibody detection, Jiangsu CDC (China) invented a COVID-19 detection box using GNPs with a diameter of 30 nm, and the test results could be obtained within 15 min [10]. In addition, GNP’s high surface-to-volume ratio also makes GNP efficient in redox catalyst materials [11,12,13]. Chen et al. [13] found that the unhindered ordered mesoporous structure of the mesoporous silica catalyst (with uniform pores of 5.6 nm) and the small size of gold nanoparticles led to high catalytic activity. In additive manufacturing, GNP’s low melting point makes it easier to sinter the product, which increases the product’s strength significantly [14,15]. The GNP’s properties, such as absorption cross-section, are sensitive to the size of the particles. Baker et al. [16] combined the bioengineering of 15 mm sericin-encapsulated gold nanoparticles (SrGNPs) with 16.63 mm levofloxacin and 17 mm barofloxacin to explore the efficiency of *Escherichia coli*, as shown in Figure 1A. The experimental results found that after adding GNPs, its activity increased exponentially with the increase of CI (using the method of Chou-T allay to calculate the synergistic, antagonistic, or additive effect of the two drugs), as shown in Figure 1B. The method of combined medication can reduce the dosage, improve the efficacy, reduce the toxicity, and reduce the occurrence of drug resistance, which is of great benefit to future treatment. Therefore, manufacturing GNPs with nano size distribution has attracted intensive attention.

Liquid laser ablation in liquid (LAL), an effective method for producing colloidal solutions of chemically pure nanoparticles, is one of the most widely utilized methods to fabricate GNPs. However, the particle size distribution produced by LAL is frequently wide, necessitating post-production fine-tuning. Laser fragmentation in liquid (LFL), which irradiates big nanoparticles in liquids with a laser, provides an ideal way to obtain GNPs with narrow size distribution. LFL is also the only approach suited for producing nanoparticles with diameters smaller than 5 nm [14]. GNPs are laser fragmented by electron absorption of laser energy, electron–phonon coupling to transmit energy to the lattice, and evaporation of superheated liquid, which results in bubbles near the GNP surface. Heat transmission between the GNPs and the liquid environment is hindered or prevented by the bubbles. The energy history of GNPs and the ensuing fragmentation and nanoparticle production processes are difficult to be characterized due to dynamic heat transfer and electron–phonon coupling.

Thereafter, it is challenging to establish a feasible method to describe laser interaction with GNPs in liquids, which can achieve efficient and high-quality preparation. In this work, molecular dynamics (MD) models, especially the cluster-based two-temperature model-molecular dynamics (TTM-MD) model, will be reviewed and discussed, which can help to observe the microscopic process GNPs’ interaction with laser light in liquid and predict the post-fabrication performance.

## 2. Molecular Dynamics Study on Laser Interaction with Bulk Material

### 2.1. Two Temperature Method Integrated with Molecular Dynamics

TTM describes electron–phonon coupling during the interaction between laser and material at the continuum level. It cannot describe the atomic level behaviors during the laser interaction with the material, such as evaporation and phase transformation. Molecular dynamics can provide an insightful view of the material, but it cannot reflect the effects of laser on the material. Ivanov and Zhigilei [17] innovatively integrate the TTM with the molecular dynamics (TTM-MD) to combine the merits of the above methods, which is a new and efficient method to describe the laser interaction with bulk material. The following equations describe a typical TTM-MD model,
(1)$$C_{e}(T_{e})\frac{\partial T_{e}}{\partial t}=\nabla \left[K_{e}\left(T_{e}\right)\nabla T_{e}\right]-G\left(T_{e}-T_{l}\right)+S\left(z,t\right),\;TTM$$
(2)$$C_{l}(T_{l})\frac{\partial T_{l}}{\partial t}=\nabla \left[K_{l}\left(T_{l}\right)\nabla T_{l}\right]+G\left(T_{e}-T_{l}\right),\;TTM$$
(3)$$m_{i}\frac{d^{2}r_{i}}{dt^{2}}=F_{i}+\frac{\frac{1}{n}\sum_{k=1}^{n}GV_{n}\left(T_{e}-T_{l}\right)}{\sum_{i}m_{i}{\left(v_{i}^{T}\right)}^{2}}m_{i}v_{i}^{T},\;MD$$
where *C* is the heat capacity, *K* is the thermal conductivity, *G* is the electron–phonon coupling constant, and *S* is the heat source from the laser. The first equation of TTM describes the electron energy evolution, which consists of the increase of the electron energy $C_{e}(T_{e})\frac{\partial T_{e}}{\partial t}$, input energy from laser $S\left(z,t\right)$, energy change by the electron heat transfer $\nabla \left[K_{e}\left(T_{e}\right)\nabla T_{e}\right]$ and electron–phonon coupling $G\left(T_{e}-T_{l}\right)$. The second equation of TTM describes how the energy of the lattice is evolved by coupling. The continuum Equation (2) is realized by adding the force term in MD during the acceleration calculation in Equation (3). The system’s total energy consisting of electron energy and lattice energy will be conserved. Specifically, for a cubic cell with the volume of V_N_, the energy transferred from the electron will be averaged within the cell and increase the force of every atom. Therefore, the whole system needs to be divided into multiple cubic cells to implement the TTM-MD coupling. However, when the number of atoms inside the cubic cell is low, the cell is assumed to be vaporized, and the coupling will stop. Since the bulk system can be easily divided by cubic cells, TTM-MD is an efficient and accurate method to describe the laser interaction with bulk material.

### 2.2. TTM-MD on Laser Ablation in Liquid and LIPSS

TTM-MD has succeeded in describing many laser processes, which were almost impossible to observe in experiments, and many mechanisms are also revealed. TTM-MD has provided an insightful view of the interaction between material and laser. One of the most important discoveries is the mechanism of the bimodal distribution of nanoparticles during laser ablation in liquid (LAL), as illustrated in Figure 2 [18]. From the snapshots, we observed the two formation channels in LAL: the first one is that the molten layer is ejected into the liquid environment and forms three large nanoparticles; the second one is that the vaporized atoms coalescence and nucleate into tiny nanoparticles. The above two formation mechanisms explain the bimodal size distribution after LAL.

In addition to laser ablation in liquid, TTM-MD has also helped discover the laser-induced periodic surface structures (LIPSS) mechanism [19], as shown in Figure 3. It was found that the liquid environment will decelerate the ablation plume and form a hot metal layer near the surface. Inside the air or vacuum, a sharper surface will be formed. The simulation reveals the complex dynamics process during the formation of LIPSS and has practical implications for nanostructure manufacturing. Shugaev Maxim et al. [20] also found the formation of LIPSS of Cr at laser fluences above the ablation threshold and helped the manufacturing speed and quality. Ivanov et al. [21] compared the TTM-MD simulation results and experiment of the same spatial and temporal scales and revealed the periodic nanostructure’s formation mechanism.

## 3. Molecular Dynamics Study on Laser Interaction with Nanoparticles

### 3.1. Cluster-Based TTM-MD for Nanoparticles

Although TTM-MD proves to be successful in laser interaction with bulk material, it cannot be applied to the nanoparticles for two reasons. First, as described in Section 2.1, one of the essential parts of TTM-MD is dividing the MD system into cubic cells to calculate the average electron and lattice temperature. Suppose this dividing method is implemented on nanoparticles. In that case, TTM-MD will mistake part of the nanoparticle as vaporized due to its low density within the cubic cell, as shown in Figure 4a,b. Such a mistake will lead to the artificial cold shell in Figure 4c, though it is assumed to be homogeneously heated by the laser in the vacuum [22]. Second, the conventional approach treats the target as bulk and assumes that the coupling equilibrium time for gold is 1 ps. However, we cannot neglect the size effect on the electron–phonon coupling when the diameter of GNP is smaller than 10 nm. The coupling equilibrium time will vary between 500 fs to 500 ps for clusters with different sizes, which leads to 1000 times coefficient magnitude difference in the coupling coefficient [23,24].

To calculate the TTM-MD individually for every nanoparticle and cluster, Hao Huang and Leonid Zhigilei proposed a new model characterized by its unique ability to consider the size effect on the electron–phonon coupling [25]. Interaction between Au is governed by the Embedded Atom Method (EAM) potential [26], which provides a realistic description of Au’s physical properties. The coarse-grained water model is implemented to describe the liquid environment [27,28], and the contact angle determines the Lennard–Johnes potential parameters between Au and water. Since the computation cost is proportional to the cubic of the radius, an advanced pressure wave transmitting spherical boundary conditions to simulate the semi-infinite liquid environment around the system was developed. TTM-MD coupling is performed within each nanoparticle/cluster instead of fixed cubic cells. The cluster size and its effect on coupling coefficient G are updated every step, but the current cluster analysis is serial, which is computationally costly and limited by the memory. Therefore, a high-performance on-the-fly parallel cluster analysis algorithm is developed to calculate the cluster size to achieve the live update for cluster size during large-scale simulation. The algorithm mainly has two stages: every processor will perform serial cluster analysis of the atoms within their domain in the first stage. Every local cluster in the domestic processor will communicate with the clusters in the neighbor processors to establish the link between the parts of one cluster in various processors. In the second stage, the cluster analysis will be performed again globally to determine the cluster size and broadcast to the whole system. In addition to the real-time cluster size update, the temporal size distribution of nanoparticles is also available, which provides a global perspective on the formation channels. More details of algorithms are illustrated in our previous paper [25].

### 3.2. Atomistic Study of Laser Fragmentation in Liquid

The cluster-based TTM-MD is firstly implemented in the study of LFL, which provides a realistic and insightful view at the atomic scale. Figure 5 shows the process of the laser interaction with nanoparticles inside the liquid from a few different perspectives, including the temperature and size evolution, bubble oscillation, and spatial distributions. Benefitting from the high performance of the cluster analysis parallel algorithm, the cluster-based TTM-MD can update the cluster size at every timestep and provides the size evolution, as shown in the bottom row of the snapshots in Figure 5. The simulation also reveals that nearly all the gold atoms have been quenched and coalesced outside the bubble. The collapse of the bubble facilitates the nucleation of the nanoparticles in the end. The simulation result agrees well with the LFL experiment results [29]. The fruitful results provided by the cluster-based TTM-MD help reveal the mechanism of the nanoparticle size distribution. They provide practical advice to narrow the size distribution by changing the laser fluence [30].

## 4. Application

This section will briefly introduce a few promising areas where the cluster-based TTM-MD can be applied, including a further and wider exploration of LFL under various conditions, nanobubble oscillation, and cancer thermal treatment by laser.

### 4.1. Laser Fragmentation in Liquid

Figure 6 demonstrates the simulation and experiments of LFL. It is a high-nonequilibrium phenomenon taking place in a nanoscale time and dimension, which involves vaporization, and nucleation in both liquid and vapor environments, as depicted in Figure 6E. Huang and Zhigilei [25] combined the classical MD method and TTM to simulate LFL processes for GNPs. As depicted in Figure 6A, the total mass of nanoparticles with various diameters by a simulation of LFL was analyzed for 20 nm gold nanoparticle, irradiated by a 10 ps laser pulse at an absorbed energy density of 2.7 eV/atom. The modeling results revealed that at 200 ps, vapor and atomic clusters smaller than 1 nm contributed to 23% of the total mass of the fragmentation product. After that moment, most metal vapors and atomic clusters were quickly injected into an aqueous environment, where they grew into nanoparticles with sizes ranging from 1 to 3 nanometers. Then, these nanoparticles contributed to 35% of the total mass of the fragmented product at 9.2 ns. Through experimental research, Werner et al. [31] found that in the LFL process, the Coulomb explosion was triggered due to possible photothermal evaporation under nanosecond laser irradiation, which led to further fragmentation of nanoparticles, as shown in Figure 6D.

Through simulation experiments, Zhigilei and Garrison [32] found that the laser energy significantly affected the fracture process of nanoparticles in their liquids, as shown in Figure 6B. For shorter laser pulses, substantial damage to the irradiated nanoparticles was observed at lower laser fluences, and the damage was characterized by distinct mechanical damage. For longer laser pulses, significantly higher laser flux was required to cause visible damage to the nanoparticles. Similarly, our results also showed that with the decrease of the laser energy, the nanoparticle would undergo a transition from a “strong” phase to a “mild” phase [25]. At lower laser radiation energies, all products of the explosive phase decomposition of the nanoparticles were rapidly injected into the water surrounding the nanobubbles formed around the nanoparticles, which resulted in the formation of a large central nanoparticle surrounded by smaller satellite fragments. In addition, Jiang et al. [33] used classical molecular dynamics and a two-temperature model to study the cavity formation process of silver nanoparticles under ultrafast laser irradiation. As shown in Figure 6C, upon laser irradiation, a small hole initially appeared in the center of the silver NPs, followed by a tiny void in them. Subsequently, a growing number of voids appeared in the silver NPs, and these small voids swelled violently, coalesced into larger voids, and hollowed out the NPs.

### 4.2. Nanobubbles

Nanobubbles (NBs) are defined as a volume of gas or vapor surrounded by a liquid with a size similar to that of nanoparticles (<1000 nm) [35]. Figure 7 demonstrates the simulation and experiments of nanobubble in LFL. As depicted in Figure 7A, the plasma cools by colliding and recombining with water molecules, causing a fast rise in water temperature and pressure, resulting in water vapor NBs surrounding the irradiated NPs [36]. Meunier et al. [37,38] were the first to use plasmon-mediated NBs, irradiating 100 nm gold nanospheres with a femtosecond pulsed laser (45 fs) and finding that the threshold for plasmonic-mediated NBs creation was similar to heat-induced NBs. Using MD simulations, Maheshwari et al. [39] investigated the formation of NBs around a heated NP in a bulk liquid. Figure 7B (a) and (b) depicted typical profiles of vapor NBs surrounding heated NPs in liquids with or without dissolved gases based on simulation findings. They calculated the average density field (as a function of time) of radial liquid particles around the NPs to explore the production of NBs. The simulation results demonstrated that, due to the system’s finite size, the radius of the NB stabilized after the initial explosive growth, as illustrated in Figure 7B (c). In addition, they compared the stable radius of NBs to the MD results (in Figure 7B (d)) by theory and found that the average prediction error was less than 10%. 

Because NBs are ideal for biological applications, such as those that can be triggered by ultrasound, they have attracted a lot of interest. For example, NBs can be used as drug carriers for intravenous injection since these tiny bubbles diffuse faster from blood vessels to surrounding tissues. At the same time, antibody functionalization on the bubbles’ surface will promote NBs’ binding to cellular targets, allowing for easier accumulation into the organization. NBs generated higher contrast in xenograft tumors compared to microbubbles (MBs) over a longer length of time [40], while having the identical mean signal intensity to MBs in vitro due to their significant capacity to enter tumor tissue. NBs and SonoVue, a commercial MB, have similar image-enhancing characteristics, according to in vitro research. The most effective approach for cancer diagnosis is to develop specific ultrasound contrast agents (UCAs) that directly target cancer. Nanobubbles enable UCAs to penetrate the endothelial space of tumor blood vessels and circulate into tumor tissue. As shown in Figure 7C, the echo of FA-NBs-IR780 was analyzed by in vitro contrast-enhanced ultrasound imaging. As expected, the echo of FA-NBs-IR780 and SonoVue MBS showed bright enhanced imaging [41]. However, some in vivo investigations have demonstrated that NBs have more tumor strength than MBs. In addition, the light-indirect responsive properties of NBs can be exploited to improve the permeability of antibiotics through biofilms [42], thereby more effectively eliminating biofilms and reducing antibiotic resistance. As shown in Figure 7D, bacterial biofilms were formed in vitro on glass surfaces (top left) for 24 h, and GNPs penetrated the biofilms (top right). After the intense nanosecond laser pulse was absorbed, NBs emerged around the AuNPs (bottom left), and the mechanical force of the NBs created more space between the cells, allowing better penetration of subsequently applied antimicrobial agents (bottom right) [43].

### 4.3. Cancer Therapy

Gold is a multifunctional substance due to its bacteriostatic, antiseptic, and antioxidant qualities. Even more sophisticated applications have been developed to use gold’s nanoscale fabrication and functionalization capabilities for targeted medication delivery to cancer locations [44]. Photothermal therapy (PTT) and photodynamic therapy (PDT) are potential cancer therapies that exogenously administer heating of nanoparticles and activation of photosensitizer (PS) medicines in response to certain wavelengths of light [32]. In terms of the PS medicines, cytotoxic photothermal heating can promote apoptotic and necrotic cancer cell death by external photoactivated light via the surface plasmon resonance (SPR) phenomenon and reactive oxygen species (ROS) [33,34,45] (in Figure 8). The SPR effect and their ease of functionalizing desired molecules using sulfur chemistry improve their ability to load PS drugs. Therefore, GNPs are the most promising photothermal and photosensitizer carriers because of their high photothermal conversion efficiency.

In the medical field, the quality and manufacturing of GNP can be improved with the help of simulation, reducing manufacturing costs. Furthermore, precision treatment will be possible to reduce or avoid the damage during PTT. Currently, the price of GNPs is relatively high, costing 1040 USD per 2.5 mg. The price is even higher when lower size dispersion is required [46]. However, by establishing a simulation model, the fabricating process, such as laser interaction with nanoparticles in liquids, can be simulated more accurately, the yield rate can be improved, and the manufacturing cost can be reduced. In addition, the model can provide precise guidance for infrared therapy, strictly control the dose of the drug, the location of injection, and evaluate the efficacy of the drug. For example, due to the reduction of manufacturing costs, GNPs can also be added to improve nucleic acid’s detection range and sensitivity [47]. For the treatment of tumor cells, moreover, the detection process of the cancer marker carcinoembryonic antigen (CEA) using GNPs is simulated in the environment of human serum, and the weak changes in the conductance process are detected to improve the detection accuracy [48].

## 5. Future and Challenges

### 5.1. Laser Interaction with Noble and Alloy Nanoparticles

Gold nanoparticles are among the most studied material because of their wide application and stability. In addition to gold nanoparticles, other noble nanoparticles, such as Pt, Ag, are also essential agents in therapy, drug delivery, and imaging [49]. LFL can also yield noble nanoparticles, and they are stable inside water. Cluster-based TTM-MD can help describe and reveal the mechanisms of LFL to other noble nanoparticles. The LFL of alloy nanoparticles, such as AuFe and AgNi, can also benefit from the cluster-based TTM-MD. The study on element preference before and after the laser interaction will be fascinating [50].

As well as fragmentation, the nanoparticle structure can be modified by the laser in liquid. For example, FeNi with the L1_0_ phase demonstrates permanent magnetic property. Alloy nanoparticles with specific structures such as core-shell can also be yielded with the help of the laser [51]. In addition to the structure, the composition of the alloy nanoparticles can also be tuned [13]. The above study on laser interaction with alloy nanoparticles can be facilitated with the help of cluster-based TTM-MD.

### 5.2. Laser Melting in Liquid of Multiple Nanoparticles

In addition to LFL, laser melting in liquid is another method to regulate the size distribution of nanoparticles. By melting nanoparticles instead of vaporizing them, larger nanoparticles will be formed. Alloy nanoparticles can also be formed by mixing colloid solutions of different elements [52], as shown in Figure 9. No numerical study has been done on LML yet. Cluster-based TTM-MD is promising to reveal the mechanisms of LML, but it will encounter the challenge of boundary conditions.

In the current TTM-MD for nanoparticles, the whole system is bordered by the non-reflecting spherical boundary condition, through which the shock wave generated from the nanoparticle will travel. Since there is only one nanoparticle, the pressure wave is assumed to be identical in every direction. However, in the case of LML, it will involve at least two nanoparticles, and the system is not spherical symmetry anymore. The pressure waves will propagate from every nanoparticle and interfere with each other. Pressure wave in a few directions is more robust and faster than others, and the spherical boundary condition will reflect them. Setting a proper boundary condition for an exotic system will be a challenge for LML in the future.

### 5.3. Ionization by Femtosecond Laser

Electron–phonon coupling and Coulomb explosion are the main mechanisms during the laser interaction with nanoparticles [53], which mainly depend on pulse duration and energy absorption. TTM-MD can only describe the electron–phonon coupling, the primary mechanism for a laser with a longer pulse. The electron–phonon coupling will heat the nanoparticle, and the mechanical shock wave is the main reason to fragment the nanoparticle. Coulomb explosion is much more violent than the pressure wave. Therefore, it plays a more critical role in femtosecond laser fragmentation. Although the ionization and the following Coulomb explosion are critical in LFL, it is challenging to observe the generation of the ions and track the violent Coulomb explosion. Cluster-based TTM-MD is a promising method to solve the problems, but in the liquid environment, the electron will form a space charge which affects the following electron emission process [54]. How to numerically describe the ionization process of nanoparticles under femtosecond laser is a challenge to implementing the cluster-based TTM-MD in the femtosecond laser fragmentation in liquid.

## 6. Conclusions

Laser interaction with nanoparticles in liquid is the fundamental theory for many applications, from cancer phototherapy and diagnosis to targeted drug delivery. However, it is a high-nonequilibrium phenomenon taking place in a nanoscale time and dimension, which involves vaporization and nucleation in both liquid and vapor environments, and only the collective behavior of colloid solution can be observed in the experiment. Hence, the mechanism of the laser interaction with the nanoparticle, especially the solo nanoparticle, is still not fully revealed. Fortunately, the continuum level TTM integrated with atomistic level MD provides a possible solution to describe the above phenomena. However, the classical TTM-MD cannot be applied in the study of nanoparticles due to its dividing method.

The cluster-based TTM-MD performs the electron–phonon coupling in each nanoparticle rather than within a fixed cubic cell and considers the size effect on coupling. A high-performance parallel cluster analysis algorithm is developed to update the cluster size and its effect on the coupling coefficient. Furthermore, the simulation results agree with the LFL experimental results. This suggests that the fruitful results provided by cluster-based TTM-MD help reveal the mechanism of nanoparticle size distribution and provide some practical suggestions for narrowing the size distribution.

In addition, the potential application of molecular dynamics for laser interaction with nanoparticles in liquids is given in this study, such as investigating the mechanism of LFL and NBs, and biological applications. Moreover, perspectives and challenges on TTM-MD for laser interaction with NPs are proposed, such as laser interaction with noble and alloy nanoparticles, laser melting in liquid of multiple nanoparticles, and ionization by a femtosecond laser, which provides a viable avenue for future research.

## Figures and Tables

**Figure 1 nanomaterials-12-01524-f001:**
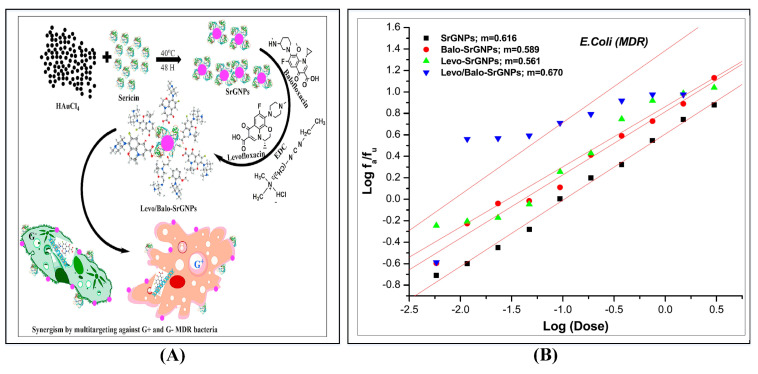
Application of SrGNPs in medicine [16]; (**A**) synergistic effect of levofloxacin, balofloxacin, and SrGNPs by targeting multi-targets; (**B**) median effect slope curve of *Escherichia coli* (MDR sclerotia). Reprinted with permission from Ref. [16]. Copyright 2020 Elsevier.

**Figure 2 nanomaterials-12-01524-f002:**
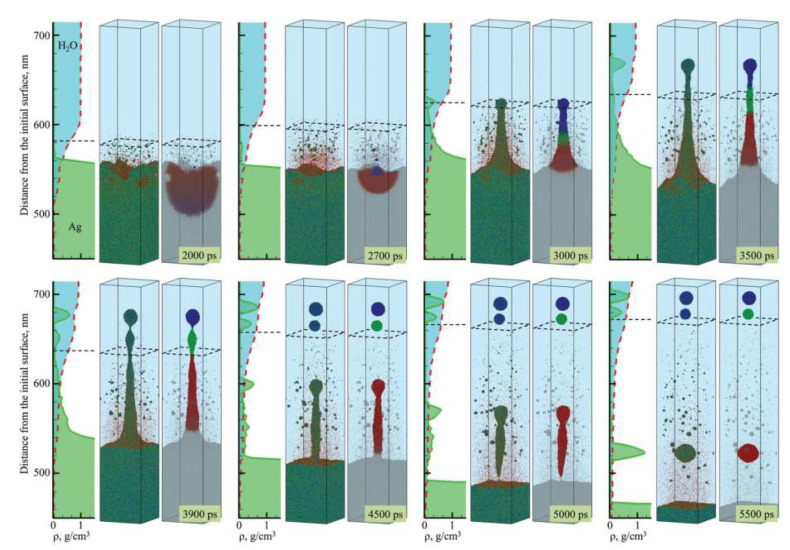
TTM-MD simulation snapshot of laser ablation in liquid [18]. Atoms are colored according to their potential energy, with blue representing crystalline nanoparticles, green representing molten silver, and red representing individual silver atoms. The degree of mercury mixing is represented by the density map, and the black dashed squares in the atomic snapshot and the horizontal dashed line in the density map indicate the approximate location of the diffusion "boundary" between the dense water and low-density mixing regions. Reprinted from Ref. [18].

**Figure 3 nanomaterials-12-01524-f003:**
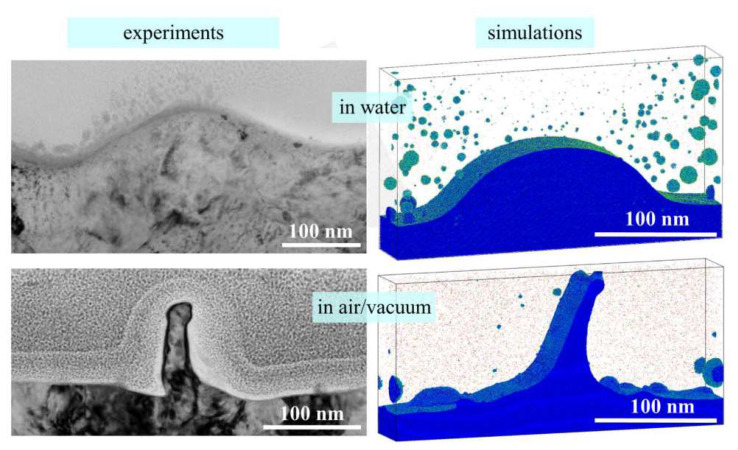
LIPSS structure comparison between the experiment and TTM-MD simulation [19]. Reprinted with permission from Ref. [19]. Copyright 2020 Royal Society of Chemistry.

**Figure 4 nanomaterials-12-01524-f004:**
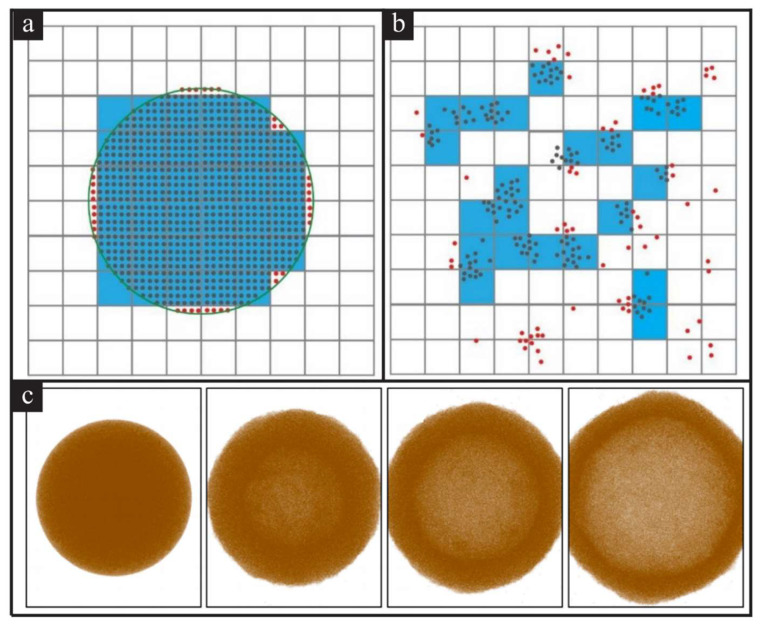
Schematic representation of the conventional approach of TTM-MD and its results. The TTM-MD calculations will be performed in cells colored blue (**a**) The initial configuration of the nanoparticle; (**b**) the atomistic distribution after fragmentation; (**c**) “cold” shell formed from homogeneously heated nanoparticle in a vacuum [22]. Reprinted with permission from Ref. [22]. Copyright 2015 American Chemical Society.

**Figure 5 nanomaterials-12-01524-f005:**
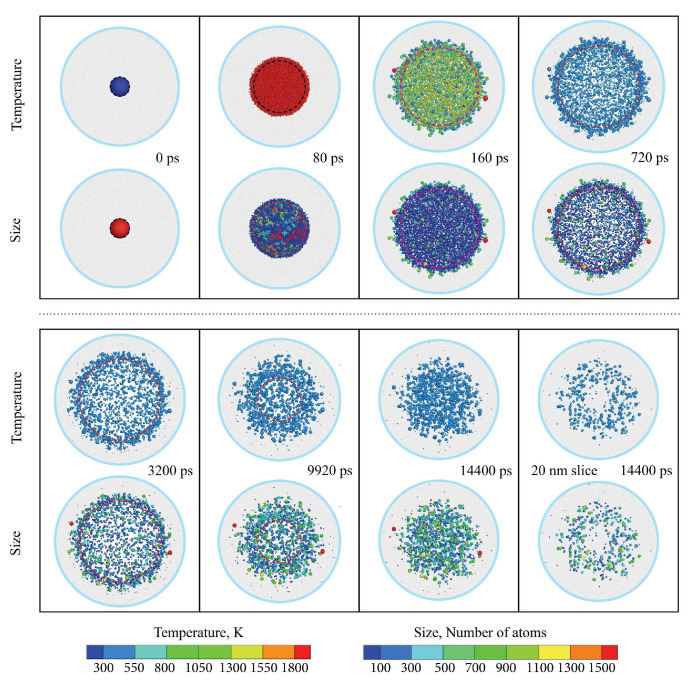
Snapshots of cluster-based TTM-MD simulation of laser fragmentation in liquid [25]. Reprinted with permission from Ref. [25]. Copyright 2021 American Chemical Society.

**Figure 6 nanomaterials-12-01524-f006:**
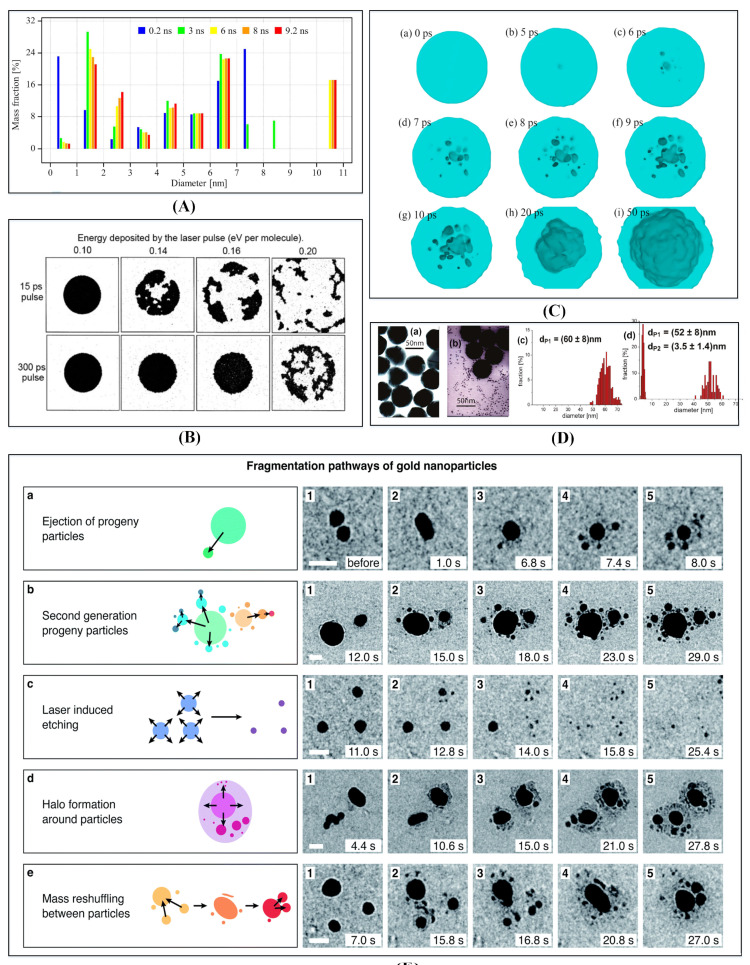
Simulation and experiments of laser fragmentation in liquid. (**A**) The total mass of nanoparticles with various diameters by a simulation of LFL of a 20 nm gold nanoparticle irradiated by a 10 ps laser pulse at a deposited energy density of 2.7 eV/atom [25]. Reprinted with permission from Ref. [25]. Copyright 2021 American Chemical Society. (**B**) Snapshots of LFL from MD simulations of laser irradiation of individual particles (~100 nm) versus deposited laser energy [32]. Reprinted with permission from Ref. [32]. Copyright 1998 Elsevier. (**C**) Simulation of the creation of hollow silver NPs using MD (20 nm silver nanoparticle after heating for 50ps) [33]. Reprinted with permission from Ref. [33]. Copyright 2021 Elsevier. (**D**) TEM images and corresponding size distributions of 60 nm GNPs before (**a**,**c**) and after (**b**,**d**) of laser irradiation (at 1 kHz and an excitation wavelength of 400 nm) [31]. Reprinted with permission from Ref. [31]. Copyright 2011, copyright American Chemical Society. (**E**) Fragmentation pathway of GNPs in LFL (under femtosecond laser) [34]. Reprinted with permission from Ref. [34]. Copyright 2021 Royal Society of Chemistry.

**Figure 7 nanomaterials-12-01524-f007:**
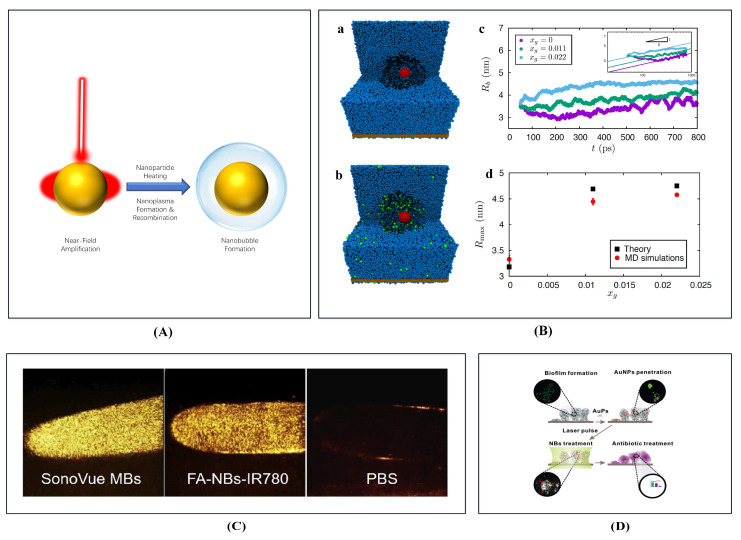
Simulation and experiments of nanobubble in LFL. (**A**) Generation of water vapor NBs around the irradiated NPs under pulsed laser irradiation [36]. Reprinted with permission from Ref. [36]. Copyright 2021 Royal Society of Chemistry. (**B**) Simulations results of NBs from MD of laser irradiation; a typical image of vapor NBs forming around heated nanoparticles in a liquid (a) and dissolved gas in the liquid (b); the radius of a NB as a function of time for various gas concentrations; (**c**), and the steady-state radius of a NB as a function of gas mole fraction in liquid molecules [39]. Reprinted with permission from Ref. [39]. Copyright 2018 American Chemical Society. (**C**) Ultrasound contrast-enhanced images of a gastric cancer xenograft with SonoVue MBs and NBs at 10, 30, 60, 180, 300, and 900 s after injection, illustrated by (**a**–**f**) and (**g**–**l**), respectively [41]. Reprinted with permission from Ref. [41]. Copyright 2019, Elsevier. (**D**) In vitro formation of bacterial biofilms on glass surfaces for 24 h through NBs [43]. Reprinted with permission from Ref. [43]. Copyright 2018 Springer Nature.

**Figure 8 nanomaterials-12-01524-f008:**
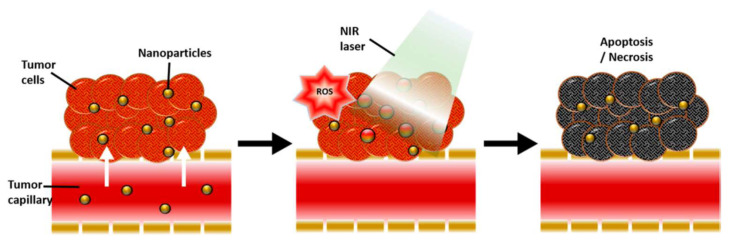
Schematic illustration of the physiological and biological effects of gold nanoparticle mediated photothermal therapy (PTT) and photodynamic therapy (PDT) [45]. Reprinted from Ref. [45].

**Figure 9 nanomaterials-12-01524-f009:**
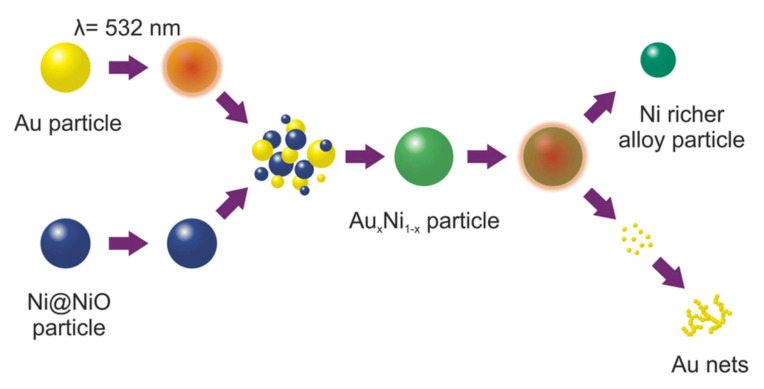
Alloy nanoparticle formation by LML after mixing different nanoparticle colloidal solutions [52]. Reprinted with permission from Ref. [52]. Copyright 2016 American Chemical Society.

## Data Availability

No new data were created or analyzed in this study. Data sharing is not applicable to this article.

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
