# Peer review of "Molecular Dynamics Study of Laser Interaction with Nanoparticles in Liquids and Its Potential Application"

_nanomaterials, 2022, doi:10.3390/nano12091524_

Round 1
Reviewer 1 Report
Dear Authors, my comments are in the attached file

Reviewer 2 Report
In the manuscript “Molecular dynamics study of laser interaction with nanoparticles in liquids and its potential application”, the authors reported the implementation of the two-temperature method integrated with molecular dynamics (TTM-MD) in laser interaction with bulk material that has shown great potential in providing a panoramic view of the laser interaction with the nanoparticles.
A new model was established to perform the TTM-MD on each individual cluster rather than the cubic cells, accompanied by the high-performance parallel cluster analysis algorithm to update the cluster size. The cluster-based TTM-MD reveals the nanoparticle formation mechanism of laser fragmentation in liquid (LFL) and helps study the effects of laser fluence on the size distribution.
The authors proved the application potential of the model to be implemented in laser thermal therapy of tumours, laser melting in liquid (LML), etc.
The result seems correct and well presented, and the methods are adequately described.
There are some minor mistakes in the text, spell check is required.
Thereby, I recommend this manuscript for publication in Nanomaterials, with minor revision.
